# *MEG3* Expression Indicates Lymph Node Metastasis and Presence of Cancer-Associated Fibroblasts in Papillary Thyroid Cancer

**DOI:** 10.3390/cells11193181

**Published:** 2022-10-10

**Authors:** Sina Dadafarin, Tomás C. Rodríguez, Michelle A. Carnazza, Raj K. Tiwari, Augustine Moscatello, Jan Geliebter

**Affiliations:** 1Department of Otolaryngology-Head and Neck Surgery, University of Washington, Seattle, WA 98195, USA; 2RNA Therapeutics Institute, University of Massachusetts Medical School, Worcester, MA 01655, USA; 3Department of Pathology, Microbiology and Immunology, New York Medical College, Valhalla, NY 10595, USA; 4Department of Otolaryngology, New York Medical College, Valhalla, NY 10595, USA

**Keywords:** thyroid cancer, long-noncoding RNA, biomarker

## Abstract

Papillary thyroid cancer is the most common endocrine malignancy, occurring at an incidence rate of 12.9 per 100,000 in the US adult population. While the overall 10-year survival of PTC nears 95%, the presence of lymph node metastasis (LNM) or capsular invasion indicates the need for extensive neck dissection with possible adjuvant radioactive iodine therapy. While imaging modalities such as ultrasound and CT are currently in use for the detection of suspicious cervical lymph nodes, their sensitivities for tumor-positive nodes are low. Therefore, advancements in preoperative detection of LNM may optimize the surgical and medical management of patients with thyroid cancer. To this end, we analyzed bulk RNA-sequencing datasets to identify candidate markers highly predictive of LNM. We identified *MEG3*, a long-noncoding RNA previously described as a tumor suppressor when expressed in malignant cells, as highly associated with LNM tissue. Furthermore, the expression of *MEG3* was highly predictive of tumor infiltration with cancer-associated fibroblasts, and single-cell RNA-sequencing data revealed the expression of *MEG3* was isolated to cancer-associated fibroblasts (CAFs) in the most aggressive form of thyroid cancers. Our findings suggest that *MEG3* expression, specifically in CAFs, is highly associated with LNM and may be a driver of aggressive disease.

## 1. Introduction

Papillary thyroid cancer (PTC) is the most common endocrine malignancy and is the primary contributor to increasing thyroid cancer incidence [1]. While most PTC subtypes are definitively treated with thyroidectomy or lobectomy alone, more aggressive phenotypes require adjuvant radioactive iodine (RAI) therapy and/or subsequent lymph node dissection. Gene expression profiling has already demonstrated the ability to characterize nodules with indeterminate cytology [2,3,4], and ongoing research aims to identify predictive molecular signatures for clinically aggressive disease [5]. Indeed, a major direction for future research is the discovery of cellular factors that serve as both prognostic markers and targets for personalized therapy [5].

Lymph node metastasis occurs in approximately 40% of adult PTC cases and is associated with higher rates of recurrence and reduced survival [6,7,8]. Current imaging modalities, including ultrasound, CT, and FDG-PET/CT, have low sensitivities for detecting tumor-positive lymph nodes in the lateral and central compartments [9,10]. Molecular markers used as adjuvants to current imaging modalities may improve the detection of metastatic lymph nodes and better guide clinical and surgical management for these patients [11]. However, the use of single-gene mutation profiles (e.g., *BRAF*) to direct surgical management toward prophylactic cervical lymph node dissection remains controversial and is not recommended in routine practice [5,12,13]. Preoperative biomarkers that can better predict lymph node metastasis may yet define the subset of PTC patients who would benefit most from node dissection.

Despite well-described effects on cellular function and oncogenesis, long-noncoding RNAs (lncRNAs) are relatively underutilized as biomarkers and therapeutic targets [14,15]. These transcripts of >200 nucleotides do not code for protein but have diverse regulatory potential in gene expression, alternative splicing, post-transcriptional mRNA modification, and epigenomic alterations [16,17,18]. Thoroughly researched lncRNAs such as *MALAT1* and *HOTAIR* [19,20,21] establish important roles in cancer biology and early prognostication. lncRNAs dysregulation in thyroid cancer [22] is associated with aggressive phenotypes [23,24] and stable enough to be detected in serum [25]. Thus, comprehensive identification of differentially expressed lncRNAs may expand our existing toolkit for PTC detection, grading, and staging.

Genome-wide investigations of patient PTCs have identified many lncRNAs that have potential diagnostic and therapeutic implications. The bulk of published PTC transcriptomic studies utilizes microarray and quantitative reverse transcription followed by PCR (qRT-PCR), revealing lncRNA dysregulation in cancerous tissue [26,27,28,29]. However, these methods screen for predetermined RNA variants and therefore probe only a fraction of the non-coding transcriptome [30,31]. More recently, RNA-Sequencing of small patient cohorts has associated select lncRNAs with molecular and clinical PTC subtypes [32,33]. The Cancer Genome Atlas Thyroid Carcinoma (TCGA THCA) project has reinforced these associations with larger datasets [24,34,35]; however, RNA-sequencing by TCGA was performed on polyA-purified RNA, which may not capture lncRNAs lacking poly-adenylated tails [36,37].

To maximize the detection of differentially expressed (DE) lncRNAs in PTC, we utilized our previously reported RNA-sequencing dataset of 44 matched-paired tumor and normal adjacent tissue samples using rRNA-depleted total RNA [38]. lncRNAs associated with lymph node metastasis were identified by examining modules of co-expressed genes that were associated with patients who had LNM. We found that *MEG3*, a lncRNA previously described as a tumor suppressor [39,40], was paradoxically highly associated with LNM and poor survival. We further investigated the cell-specific expression of *MEG3* in single-cell thyroid cancer data and found expression was nearly isolated to cancer-associated fibroblasts (CAFs) and that knockdown of *MEG3* in human fibroblasts downregulates the expression of matrix metalloproteases (MMPs) previously identified as contributors to cancer metastasis. Overall, our analysis identifies *MEG3* expression as highly associated with LNM in thyroid cancer with a potential role in contributing to metastatic potential via its expression in CAFs.

## 2. Materials and Methods

### 2.1. NYMC Patient Specimens

Existing specimens from 44 patients who underwent thyroidectomy with fresh frozen thyroid tissue were collected between 2009 and 2013. All tumors had corresponding matched normal-adjacent tissue, and the diagnosis of PTC was validated by pathological examination. RNA extraction, preparation, and sequencing are as previously described [38]. An R-based data object for the NYMC dataset is hosted (github.com/umasstr/NYMC-PTC, accessed on 8 August 2020) with information regarding the download and visualization of these data included.

### 2.2. lncRNA Annotation

Gene biotypes were obtained from GENCODE [41] and genes classified as lncRNAs had one of the following biotype annotations: 3prime_overlapping_ncRNA, antisense, bidirectional_promoter_lncRNA, lincRNA, macro_lncRNA, non_coding, processed_transcript, sense_intronic, and sense_overlapping.

### 2.3. BRAF Genotyping

*BRAF^V600E^* mutations were detected from patient samples using TaqMan probes previously described by Benlloch et al. [42]. Briefly, 100ng genomic DNA was extracted from remaining TRIzol fractions after first removing the aqueous layer containing RNA and precipitating the DNA. Forward and reverse primers, as well as mutant and wild-type probes, were designed to detect *BRAF^V600E^* and *BRAF^WT^* DNA, respectively. Primer probe sequences can be found in Appendix A. Real-time PCR was performed using the TaqMan One-Step RT-PCR Master Mix Reagents kit (Applied Biosystems, Waltham, MA, USA), and amplification and detected were performed with the ABI PRISM 7900 (Applied Biosystems, Waltham, MA, USA). BRAF mutational status was validated via manual examination of aligned RNA-Seq reads using IGV (Broad Institute, Cambridge, MA, USA).

### 2.4. Weighted Gene-Co-Expression Analysis (WGCNA)

We performed WGCNA [43] with 20,512 genes passing filtering and constructed co-expression gene networks using the optimal power 7 as determined by the scale-free topology criterion [44] and a minimum of 20 genes per module. Nineteen modules of co-expressed genes were constructed with a range of 33 (light yellow) to 5992 (turquoise) genes. Gene sets within each module were subject to MSigDB Hallmark Gene Set Enrichment Analysis [45] to identify biological processes common to co-expressed genes.

### 2.5. Fusion Detection

Fusions events were detected and annotated using a combination of STAR alignments and the STAR-SEQR tool (https://github.com/ExpressionAnalysis/STAR-SEQR, accessed on 12 June 2019). STAR-SEQR hits were filtered with the following parameters: (1) Fusion genes are only present in PTC samples; (2) at least 5 reads overlapping the cross junction must be present; (3) fused genes must be translocated from different chromosomes, or genomic distance between the genes must be >150 kb. RT-PCR followed by gel electrophoresis was performed to validate filtered hits. Fusion details and primer sequences used are provided in Appendix A.

### 2.6. TCGA Data

Level three RNA-Seq data were downloaded from the UCSC Xena Browser [46].

### 2.7. Pathway and GO Enrichment Analysis

KEGG Pathway and Gene Ontology enrichment analyses were performed on Advaita’s iPathwayGuide (http://www.advaitabio.com/ipathwayguide, accessed on 6 June 2019) platform using DE genes (abs(log2FC) > 1.5 and q-value < 0.05). Statistical tests of pathway and GO term enrichment were adjusted using FDR correction.

### 2.8. Thyroid Differentiation Score and ERK Signature

We calculated the thyroid differentiation score using 16 thyroid metabolism and function genes first characterized by TCGA THCA study [47].
TDS=Average[Log2(FC)] across 16 genes

MAPK signaling activity was measured using the ERK signature score from TCGA using 52 MAPK signaling genes first described by Pratilas et al. [48].
$$ERK\;score=log2\sum_{n=1}^{52}[\left(median(RSEM_{n}all\;patients\right)-(RSEM_{n}individual\;patient)]/52\;$$

### 2.9. Statistical Analysis

Hierarchical clustering, heatmap generation, Spearman, and Pearson correlation analyses of DEGs were performed on R version 3.5.3. Pearson’s chi-square test or Fisher’s exact test (when samples were <5) was used to analyze categorical variables. The R package pheatmap (http://rpackages.ianhowson.com/cran/pheatmap/, accessed 10 January 2021) was used for heatmap generation and hierarchical clustering.

### 2.10. Estimating Tumor Infiltration with CAFs Using TIMER2.0

We used TIMER2.0 [49] to investigate the correlation between *MEG3* expression and infiltration of CAFs in the TCGA THCA dataset. “Purity Adjustment” option was selected to account for the confounding effect of tumor purity and immune cell type infiltration.

### 2.11. Single-Cell Anaplastic Thyroid Cancer Data

Single-cell data from 5 previously described anaplastic thyroid cancer samples [50] were analyzed using the CancerSCEM webtool [51].

## 3. Results

### 3.1. Clinical Characteristics of PTC Cohort

We analyzed transcriptomic data from 44 PTC samples with matched normal adjacent tissue (NAT) collected from patients undergoing thyroidectomy between 2009 and 2013 as part of the NYMC dataset [38]. Most patients in this cohort had indicators of aggressive disease, including high rates of *BRAF^V600E^* mutation (80%), capsular invasion (77.8%), multifocality (88.9%), and higher T stage (75.6% T3 or higher) (Table 1). Previous studies of PTC epidemiology report *BRAF^V600E^* mutations present in 40–60% of PTC cases [52,53], although after reclassification separating the follicular variant PTC (FVPTC) from classical PTC (cPTC), Yoo et al. found 71.4% of cPTC harbored BRAF mutations and more aggressive phenotypes compared to other subtypes [54]. Clinicopathological characteristics of the NYMC dataset showed no significant difference in age, tumor size, lymph node metastasis, or extracapsular invasion when comparing *BRAF^V600E^* and *BRAF^WT^* tumors, although statistical analysis may be impacted by the limited number of *BRAF^WT^* tumors.

### 3.2. Analysis of Tumor vs. Normal Transcriptomics

Principal component analysis (PCA) on all specimens demonstrated distinct separation on the PC2 axis between PTC and normal adjacent tissue (Figure 1A). Next, we measured thyroid cell differentiation and activation of the MAPK signaling pathway using scoring methods developed by TCGA: the thyroid differentiation score (TDS) and ERK score (Figure 1C) [47]. Consistent with results from the TCGA and Song et al. 2018 [55], *BRAF^V600E^*-mutant PTC scored lower on the TDS compared to *BRAF^WT^* tumors (*p* = 4.5 × 10^−6^). Furthermore, *BRAF^V600E^* mutant samples displayed varying TDS scores ranging from −0.48 to −3.48, representing high and low differentiation, respectively, while the TDS of *BRAF^WT^* tumors ranged from –0.029 to –1.81. The ERK score, which represents no clear distinction was made between male and female tumors based on BRAF status, ERK score, or TDS (Figure 1C). KEGG pathway enrichment analysis of differentially expressed (DE) genes identified cell adhesion molecules and ECM-receptor interaction as well as pathways related to host immune function (Cytokine–cytokine receptor interactions and allograft rejection) as the most highly enriched among the NYMC dataset (Figure 1D). These findings are consistent with those by Song and colleagues that showed cell adhesion molecules and ECM-receptor interaction pathways were enriched among upregulated genes in PTC [55].

Four *BRAF^WT^* PTC tumors harbored fusion genes: *CCDC6-RET*, *TRIM27-RET*, *ACBD5-RET,* and *TPM3-NTRK1*. Validated fusion genes, breakpoint regions, and expression levels are available in Appendix A. All four fusion genes were previously reported in PTC [47,54,56,57].

### 3.3. WGCNA Identifies a Gene Co-Expression Module Associated with LNM

lncRNAs play a key role in transcriptional regulation of protein-coding genes [58], though often in complex genetic circuits not easily reconciled with clinical phenotypes. Our analysis of the NYMC dataset identified 756 non-coding RNAs that underwent ≥1.5-fold change between tumor and normal (q-value ≤ 0.05) (Figure 1E). To identify lncRNAs related to available sample characteristics, we constructed 19 gene network modules from pairwise expression correlations of 20,512 coding and non-coding genes using WCGNA. Modules underwent MSigDB Hallmark Gene Set enrichment analysis, and their eigengenes (the first principal component of each module) were associated with clinicopathological features (Figure 2A). Among the clinical traits probed, lymph node metastasis (LNM) showed the strongest association to a gene module (black, *p* = 0.05). In total, 188 genes were upregulated and 482 were downregulated in the black module and hallmark pathways enriched included epithelial–mesenchymal transition (*p* = 1.33 × 10^−81^), TNFα Upregulation (*p* = 2.03 × 10^−43^), and Hypoxia (*p* = 1.82 × 10^−17^) (Figure 2B). Several genes that were previously shown to contribute to aggressive phenotypes in thyroid cancer, including SERPINE1 and TBX15 [59,60], were identified in this module and significantly correlated with LNM (Appendix A). We did not appreciate a gene cluster among black modules genes that would distinguish LNM+ and LNM- tumors (Figure 2C).

Among the 74 lncRNAs co-expressed within the black (LNM) module, we examined those with the largest Gene Significance score (GS) (Table 2). Among these lncRNAs, *LINC00346*, *CASC15, EGFR*-*AS1*, *DIO3OS,* and *LINC00702* were all previously reported to contribute to proliferation and invasion [61,62,63,64]. Interestingly, we identified *NBAT1,* a tumor suppressor gene in other cancer types [65], in the LNM module. Five lncRNAs were differentially expressed only in tumors with LNM; however, the fold change between tumor and normal was only significantly greater among LNM-positive tumors for *NBAT1, RP11-815J21.4,* and *RP11-106D4* (Figure 2D).

Ranked by module membership (MM) score denoting correlation between gene expression and eigengene, black module constituent, *MEG3,* was most predictive of LNM. Kaplan–Meyer analysis of *MEG3* in the TCGA shows lower overall survival in patients with higher expression of *MEG3*; this effect intensifies in BRAF mutant patients (Figure 3A). We also examined TCGA PTC transcriptomes and found that *MEG3* is expressed higher in metastatic tumors compared to normal, though the former is poorly represented in this study (*n* = 8) (Figure 3B). Using qPCR, Wang et al. [66] found *MEG3* to be downregulated in LNM-positive PTC patient samples and that overexpression of *MEG3* reduces migration and invasion in vitro via *RAC1* inhibition. In contrast, our RNA-seq data show that *MEG3* is downregulated in LNM-negative samples (q-value = 0.0045) and expression is unchanged in LMN-positives (q-value = 0.86) (Figure 3C). We hypothesized that qPCR transcript quantification may not capture the full library of *MEG3* lncRNA variants expressed in PTC. To reconcile discordant trends in tumor *MEG3* expression obtained by qPCR [66] and RNA-Seq, we surveyed all *MEG3* isoforms found in at least 50% of samples in our dataset. Five isoforms (ENST00000455531, ENST00000398460, ENST00000522771, ENST00000398461 and ENST00000452120) passed filtering criteria. ENST00000452120 displayed the highest overall expression in tumor and normal tissue, was significantly downregulated in LMN-negative tumors, and showed higher (non-significant, *p* = 0.2) expression in LMN-positive tumors. No obvious isoform switch of the *MEG3* gene was observed between LNM-positive, LNM-negative, and normal samples (Figure 4A). Furthermore, Wang et al. qPCR probes showed no enrichment bias toward any subset of *MEG3* isoforms (Appendix A).

The study by Wang et al. used immortalized PTC cell lines in monoculture to study the role of *MEG3* in invasion and metastasis. A potential reason for the discrepancy between that study’s results and our findings from the TCGA and NYMC dataset is that *MEG3* plays unique roles in cell-autonomous mechanisms vs. its contribution to the tumor microenvironment. Tumors often recruit or modify members of the microenvironment, such as infiltrating immune cells, vascular endothelial and smooth muscle cells, and fibroblasts, to promote metastatic progression [67,68]. We therefore used TIMER2.0 to investigate the correlation between *MEG3* expression in the TCGA dataset and infiltration across cell types in the tumor microenvironment. Cancer-associated fibroblasts were strongly associated with *MEG3* expression across all four deconvolution algorithms that can measure this cell type (Figure 4B). Given this high correlation, we hypothesized that *MEG3*-expressing tumors may be secreting cytokines that recruit and activate tumor fibroblasts, including members of the TGFβ superfamily and PDGFα/β. Therefore, we calculated the Pearson correlation between *MEG3* expression and these cytokines in our tumor samples and found TGFβ3 to be the most significantly associated with *MEG3* expression (Pearson’s rho = 0.74).

To determine the cell type within the thyroid cancer tumor environment that expresses *MEG3*, we aimed to use existing single-cell RNA-sequencing (scRNA-Seq) data from thyroid cancer samples with high metastatic and invasive potential. We therefore examined five anaplastic thyroid cancer samples, given they are the most aggressive form of thyroid cancer that frequently originates from the context of dedifferentiating PTC [69]. We found that four out of five samples showed that expression of *MEG3* was nearly isolated to CAFs (Figure 5). Furthermore, the only sample lacking *MEG3* expression had the highest cellular purity as scored, suggesting the single-cell suspension of this sample may be enriched with malignant cells and therefore lack sufficient CAFs to detect *MEG3* expression (Appendix A).

From the single-cell data we examined, there was clear heterogeneity in the expression of *MEG3* among tumor-associated fibroblasts. To better understand functionally how *MEG3*-expressing fibroblasts differed from those lacking *MEG3* expression, we examined gene expression data of human fibroblasts with *MEG3* experimentally downregulated. We examined published data from Mondal and colleagues who performed RNA microarrays on human fibroblasts treated with *MEG3* siRNAs [70]. Among genes significantly downregulated by *MEG3* knockdown, as determined by a −2 fold-change from *MEG3*-knockdown to control, were MMP-1, MMP-9, and MMP-16, three metalloproteases with previously characterized roles in tumor metastasis [71,72,73].

## 4. Discussion

Despite a generally favorable prognosis, papillary thyroid cancer can transition into aggressive subtypes and metastatic disease in select patients. A toolkit of pharmaceutical, surgical, and targeted radiation-based therapies has extended longevity of these individuals; however, it remains unclear which patients will benefit most from a given intervention in the setting of advanced PTC. As molecular profiling technologies—most prominently high-throughput sequencing—become cheaper and more accessible, novel genetic hallmarks in PTC could inform the clinical approach. In particular, the detection of LNM in the central and lateral neck compartments in the preoperative setting potentially alters the surgical management of patients [74,75].

Here, we analyzed a large set of PTC transcriptomes with a focus on lncRNAs, given their increasingly recognized role in a variety of cancer-related processes yet underutilization as biomarkers relative to protein-coding genes [17,76,77]. We sought to enrich lncRNAs tied to aggressive features of PTC. Using WGCNA, we identified a module of 729 genes associated with epithelial–mesenchymal transition that exhibits a strong correlation to LNM-positivity. Most notably, the lncRNA *MEG3* was determined to have the highest module membership score, indicating it may be highly interconnected within this co-expression module as a potential hub gene. We further found increased *MEG3* expression in PTC to be associated with LNM, worse overall survival, and higher infiltration with CAFs.

*MEG3* has been described as a tumor suppressor in many cancer types, including glioma, hepatocellular carcinoma, and even thyroid cancer [39,66,78,79]. However, these studies relied on the expression of *MEG3* in tumor cell lines, bulk tissue with predominantly malignant cells, and animal models derived from xenografts of primary tumors. More recent investigations of the tumor microenvironment via single-cell technologies have started to show a diverse role of *MEG3* depending on its cellular context. Pan and colleagues demonstrated using scRNA-seq of pancreatic ductal carcinomas that *MEG3* expression was significantly enriched in CAFs of primary tumors, and *MEG3* expression was significantly associated with epithelial-mesenchymal transition signatures [80]. Given our finding that the proportion of CAFs was highly associated with *MEG3* expression in our bulk tumor samples, we aimed to determine if the *MEG3* was being expressed by tumor cells that were recruiting CAFs, or if CAFs were directly expressing *MEG3*. To determine the cell types that express *MEG3* in aggressive thyroid cancer, we analyzed existing scRNA-seq data from anaplastic thyroid cancer. Four out of five samples demonstrate *MEG3* expression in CAFs but little expression in tumor cells, suggesting CAFs are the primary cell types expressing *MEG3*.

We reasoned that the presence of *MEG3*-expressing CAFs drives metastatic potential through reorganization of the extracellular matrix to drive tumor invasion. We therefore examined gene expression data for *MEG3*-knockdown in human fibroblasts. MMP-1, MMP-9, and MMP-16 were downregulated in *MEG3*-knockdown fibroblasts, suggesting *MEG3* regulates the expression of these metalloproteases with implications in CAF-driven metastasis. Mondal and colleagues showed that *MEG3* regulates gene expression by targeting chromatin regions and forming RNA-DNA triplex structures at distal regulatory elements and recruiting polycomb repressive complex 2 (PRC2) to inhibit expression [70]. Interestingly, *MEG3* was shown to downregulate TGFβ receptor 1 in fibroblasts by Mondal, which may appear discordant with our finding CAFs expressing *MEG3* are associated with LNM-positive thyroid cancer since TGFβ signaling is needed for fibroblast activation. However, one possibility is that *MEG3* expression may be downstream of TGF B-signaling and a component of the negative feedback loop of the TGFβ signaling cascade [81]. Furthermore, Terashima and colleagues demonstrated that *MEG3* knockdown inhibits TGFβ-induced EMT in lung cancer cell lines [82]. A study of *MEG3* expression in idiopathic pulmonary fibrosis using scRNA-seq demonstrated that *MEG3* regulates the differentiation of bronchial cells through diverse mechanisms, including through *YAP1, NOTCH,* and *SOX2* signaling [83]. Future functional studies are needed to further determine the biological role of *MEG3* in CAF activation and its potential contribution to invasion and metastasis.

A key limitation of our study is that normal adjacent tissue while carrying histologic features such as healthy tissue have distinct gene expression patterns and may represent an intermediate state or the effects of field cancerization [83]. A recent analysis of adjacent tissue and healthy donor tissue across eight tumor types, including thyroid cancer, found a majority of differentially expressed genes between tumor and adjacent tissue overlaps with gene differentially expressed between tumors and healthy donor tissue, and a limited number of genes showed discordant patterns of differential expression [84]. Nevertheless, future studies should validate coding and non-coding transcripts that we identified in this study by comparing healthy thyroid tissue to PTC.

In conclusion, we utilized multiple transcriptomic datasets with matched-paired controls for the identification of lncRNA in a gene co-expression network associated with LNM and enriched in epithelial–mesenchymal transition gene set. The lncRNA *MEG3* was identified as a hub of this co-expression module and an indicator of LNM and CAFs infiltration. The identification of *MEG3*-expressing CAFs rather than malignant cells via scRNA-seq suggests this lncRNA may be playing unique roles in tumorigenesis, invasion, and metastasis based on the cellular context. However, further investigation is needed to fully elucidate the diverse mechanisms *MEG3* has and its potential use in personalizing management for PTC.

## Figures and Tables

**Figure 1 cells-11-03181-f001:**
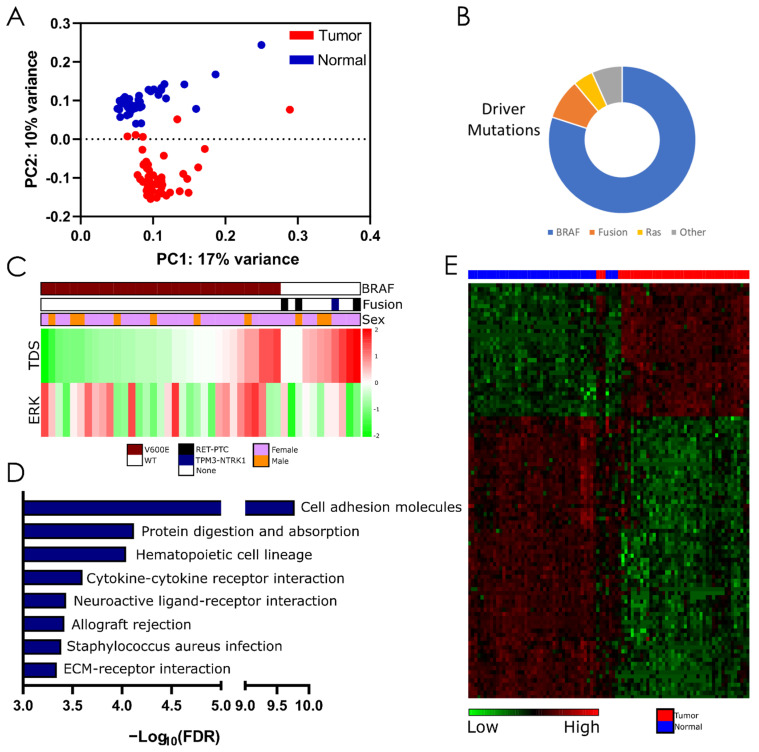
Differential expression analysis between tumor and normal thyroid tissue. (**A**) Principal component analysis of 44 PTC and matched normal tissue. (**B**) Proportion of known driver mutations identified in tumor samples. (**C**) Mutational status, sex, TDS, and ERK score in individual tumors. TDS and ERK scale represent Log2-scaled high and low thyroid differentiation and ERK-activation based on gene signatures [47]. (**D**) Pathway analysis of differentially expressed genes. (**E**) Hierarchical clustering of the top 100 differentially expressed lncRNAs (scaled normalized reads by column). FDR, false discovery rate; TDS, thyroid differentiation score.

**Figure 2 cells-11-03181-f002:**
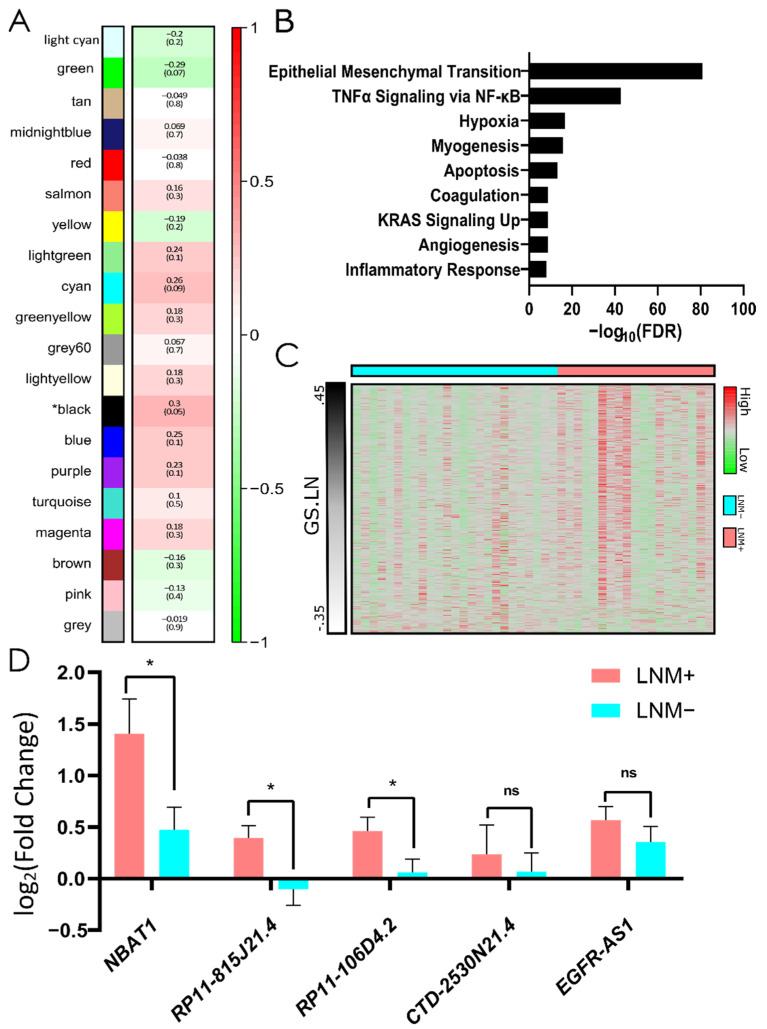
LNM-related co-expression analysis of PTC samples. (**A**) Relationship between WCGNA modules and LNM status. Values represent Pearson correlation coefficient and the correlation *p*-value (in brackets) between LNM and module eigengene. (**B**) Hallmark gene set analysis of the module most correlated with LNM status (black). (**C**) Black module constituent gene expression ranked by gene significance value related to lymph node metastasis (GS.LN) (**D**) Black module lncRNAs upregulated in LNM+ tissue but not LNM- tissue. LNM+ and LNM−, positive and negative lymph node status. * *p* < 0.05; ns, non-significant.

**Figure 3 cells-11-03181-f003:**
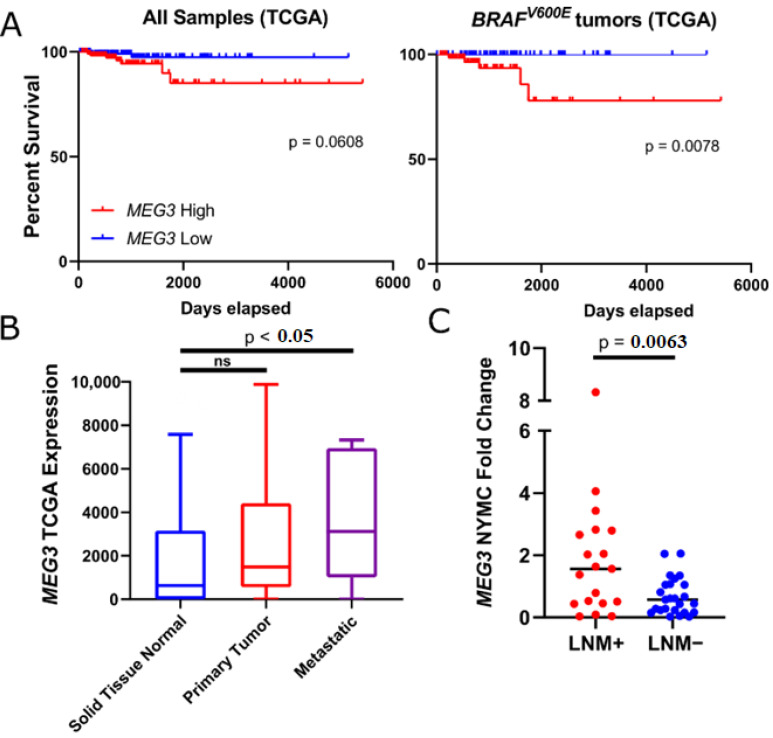
*MEG3* expression is increased in PTC with LNM and correlates with poor outcomes (**A**) TCGA all patients (left) and *BRAF^V600E^* cohort (right) survival given top (red) and bottom (blue) quartile *MEG3* expression. (**B**) TCGA Solid Tissue Normal *MEG3* expression. (**C**) *MEG3* fold-change expression in NYMC LNM+ and LNM-PTC.

**Figure 4 cells-11-03181-f004:**
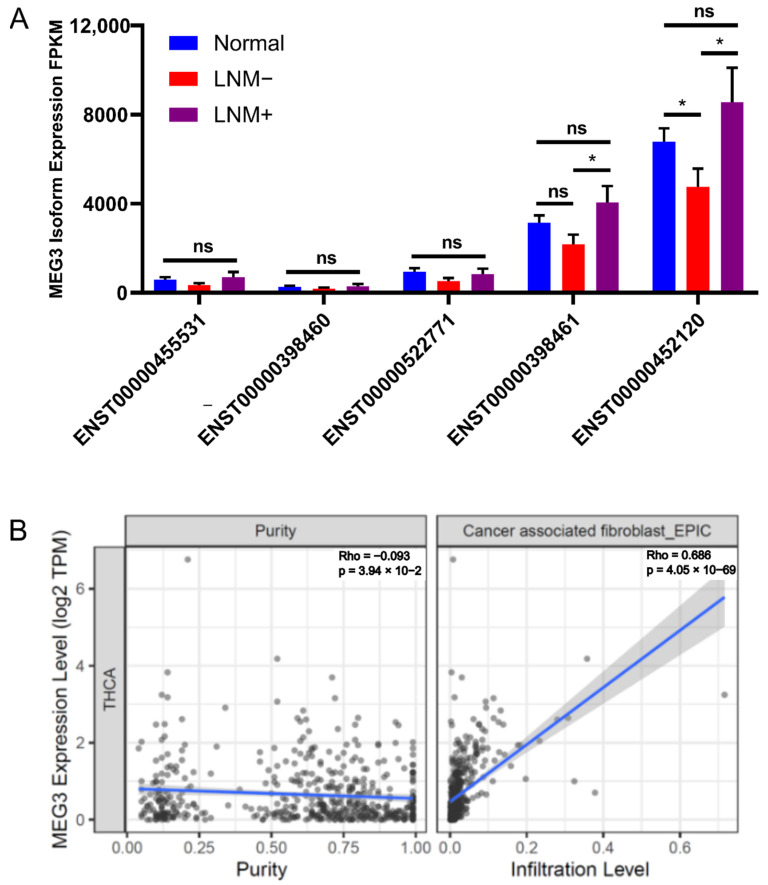
*MEG3* isoform expression and association with infiltration of CAFs. (**A**) *MEG3* transcript isoform usage in normal (non-cancerous) tissue and NYMC PTC (LNM+/−). (**B**) Scatter plots demonstrating the correlation of *MEG3* expression in TCGA THCA project with tumor purity and estimated infiltration level of CAFs using TIMER2.0. EPIC output was used to represent fibroblast infiltration estimates from TIMER2.0. * *p* < 0.05; ns, non-significant.

**Figure 5 cells-11-03181-f005:**
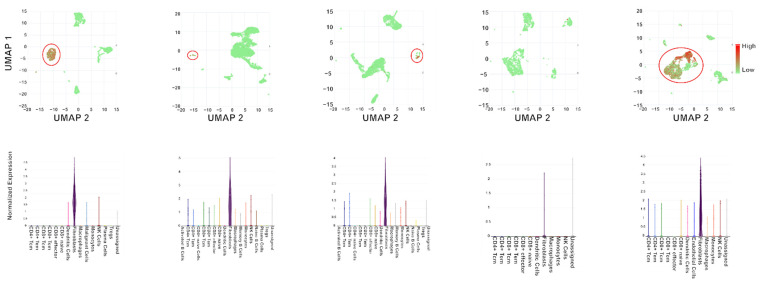
Single-cell RNA-sequencing results identify cell types that express *MEG3*. Expression profile of *MEG3* in 5 ATC patient tumors represented as UMAPs with overlaid heatmaps demonstrating *MEG3* expression and expression in different cell subtypes in each sample. Red circles represent clusters of cells that are annotated as fibroblasts.

**Table 1 cells-11-03181-t001:** Patient demographic and clinicopathological characteristics.

	BRAFV600E	BRAFWT	Total	*p*-Value
N	36 (80%)	9 (20%)	44	
Age	47.8 (20–76)	49 (34–60)	48 (20–76)	0.41
Female sex	26 (72%)	8 (89%)	34 (76%)	0.65
Thyroiditis	17 (47%)	5 (56%)	22 (49%)	0.72
Tumor size	1.9 cm (0.45–4.3 cm)	1.7 cm (0.9–3.2 cm)	2.8 cm (0.45–4.3 cm)	0.54
Lymph node metastasis	15 (42%)	4 (44%)	19 (42%)	1
Invasion	27 (75%)	7 (77%)	34 (76%)	1
T stage				1
T1–T2	9 (25%)	2 (22%)	11 (24%)	
T3–T4	27 (75%)	7 (77%)	34 (76%)	

**Table 2 cells-11-03181-t002:** Top 15 differentially expressed black module lncRNAs.

Gene Name	GS.LN	p*.GS.LN	MM.black	p.MM.black
LINC00346	0.395204	0.009589	0.623782	1.02 × 10^−05^
RP1-79C4.4	0.383766	0.01211	0.760822	4.99 × 10^−09^
RP5-1071N3.1	0.381124	0.012767	0.306905	0.048048
AC159540.2	0.31683	0.040921	0.290224	0.062257
MEG3	0.316771	0.040961	0.91168	4.85 × 10^−17^
NBAT1	0.308945	0.046506	0.682119	6.54 × 10^−07^
RP11-124N14.3	0.30859	0.046772	0.502946	0.000687
ITPK1-AS1	0.29417	0.058626	0.362727	0.018234
CASC15	0.287496	0.06487	0.68056	7.09 × 10^−07^
EGFR-AS1	0.267027	0.087358	0.213671	0.174238
DIO3OS	0.262486	0.093083	0.700355	2.43 × 10^−07^
DNM3OS	0.26201	0.093699	0.588983	4.07 × 10^−05^
LINC00702	0.257338	0.099919	0.66197	1.81 × 10^−06^
TEX26-AS1	0.254926	0.10325	0.690438	4.20 × 10^−07^
RP11-273B20.1	0.25004	0.11026	0.585077	4.71 × 10^−05^

GS, gene significance; MM, module membership, which indicates Pearson correlation between gene expression and module eigengene. * *p*. denotes correlation coefficient *p*-value.

## Data Availability

Level three RNA-Seq data were downloaded from the UCSC Xena Browser [46]. Single-cell RNA-seq data were obtained using the CancerSCEM webtool [51].

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
