# Peer review of "MEG3 Expression Indicates Lymph Node Metastasis and Presence of Cancer-Associated Fibroblasts in Papillary Thyroid Cancer"

_cells, 2022, doi:10.3390/cells11193181_

Round 1

Reviewer 1 Report

In the manuscript, the authors investigated lymph node metastasis (LNM) in papillary thyroid cancer patients, and identified multiple lncRNAs to be potentially associated with LNM. Notably, using TCGA and additional validation, they showed that MEG3 has the highest potential to be associated with LNM. Overall, the paper was well-written, and the conclusions can be sufficiently supported by the proposed methods and results. I have some minor comments as listed below:

1.       Lines 132 and 135, the formulas were not clear enough. I would suggest using standard mathematical notations. Please also add a brief explanation of both TDS and ERK scores, such as their scale, interpretation and how they are associated with PTC status/severity.

2.       Typo in Table 1, 19 should correspond to about 42% of all 45 samples.

3.       Line 176, the reference to Fig. 1B seems to be Fig. 1C.

4.       Please add additional clarification for Fig. 2C. Currently, it is not clear what the plot tries to show. Maybe add some additional labels for those candidate lncRNAs.

Author Response

We thank the reviewer for their consideration and input. Please find our responses to the reviewer suggestions below:

  1. We’ve made changes to our equations to clarify the calculation we made fot eh TDS and ERK scores. The TDS score equation has not been significantly changed as we with to keep it as closely aligned to the original source from the TGCA. Brief explanations of the scores have now been incorporated into the main text in lines 186-187.
  2. Thank you for pointing out this error, the typo in Table 1 has been corrected
  3. The reference in line 176 has been corrected
  4. We’ve added clarification for Fig 2C plot in the main text in lines 211-212, namely that we did not identify a gene cluster that would adequately distinguish LNM+ from LNM- tumors.

Reviewer 2 Report

The manuscript by Dadafarin and colleagues titled 'MEG3 expression indicates lymph node metastasis and presence of cancer-associated fibroblasts in papillary thyroid cancer' is an innovative manuscript that makes use of publicly available datasets to identify the lncRNA MEG3 as a potential novel biomarker of aggressive PTC. 

The authors designed their analysis very thoughtfully and took into account several issues that generally lack in the literature, e.g. MEG3 isoforms, etc.

There are only minor amendments I would recommend: 

1) In the discussion section, there are more papers connecting MEG3 with TGFβ signaling (one that comes to mind has to do with pulmonary fibrosis, which could be indirectly relevant to this study).

2) The performed analysis is an exemplar of analyses that can be performed with public data, so I would highly encourage the authors to share their code scripts in a public repository, such as GitHub.

Author Response

We thank the reviewer for their consideration and input. Please find our responses to the reviewer suggestions below:

  1. We’ve incorporated further evidence from the literature of MEG3’s influence of TFGb signaling pathways in our discussion. We also included a discussion of alternate pathways shown to be regulated by MEG3 via a publication examining its role in idiopathic pulmonary fibrosis (lines 358-362)
  2. We have included a github page for our analysis of the NYMC dataset that includes instructions on how to access the data (line 117). For other publicly available datasets (e.g. TCGA), we used webtools from UCSC Xena and TIMER which we will defer to their respective websites for instructions on using their suite of analysis tools.